



# Optimal site selection for Choutuppal geomagnetic observatory, based on geophysical evidences

Divyanshu Dwivedi[1], Sneh Yadav[2], Kusumita Arora[1], Rakesh Murteli[1], Alok Taori[3]

[1]Geomagnetism, CSIR-National Geophysical Research Institute, India
[2]Kurukshetra University, India
[3]National Remote Sensing Centre, India
Correspondence: ddwivedigp@gmail.com

**Abstract**

The development of the stages of the Choutuppal magnetic observatory over last 15 years has enabled the effects of the natural environment like groundwater changes and lightning activity on the magnetic data to be evaluated. A new survey for total field anomalies and analysis of lightning data is carried out to understand the nature of the subsurface. Based on model from high resolution magnetic data and conductivity depth slices from ERT and EVRI surveys, the distribution of sandy regolith, saprolite, and granitic layers in the shallow subsurface to be delineated. This model provides information for selecting a location to install the magnetic observatory by taking into account topography, lightning effect, soil resistivity, low magnetic gradients, and distance from the recharge pond.

**Keywords:** Choutuppal observatory, Magnetic anomaly, Spectral analysis, Lightning data



## 1. Introduction:

The Choutuppal (CPL) geo-electric observatory (Geographic coordinates: 78.920E, 17.290N; Geomagnetic coordinates: 149.24E, 7.47N) of CSIR-NGRI was established near Choutuppal town in the Nalgonda district, approximately 60 km southeast of Hyderabad city, Telangana state (Sanker Narayan, 1964). The region primarily comprises granite and gneissic formations. These rocks are part of the Peninsular Gneissic Complex, which is one of the oldest geological formations in India, dating back to the Archean era. The weathering of the granitic and gneissic rocks has led to the formation of red and lateritic soils. The granitic formation is encroached locally by discontinuities such as dikes or quartz reefs but these are not present on the site (Guiheneuf et al., 2014). The area around the CPL observatory mainly consists of alkali feldspar granite (Figure 1a). The regional geology of resistive granitic basement rocks, uniform soil cover, arid vegetation, and gentle topography for effective drainage of runoff water during rainy seasons were assessed to be suitable for geo-electric measurements (Sankar Narayan et al., 1967; Sarma et al., 1969). Below the surface, shallow drillings reveal: 1) A sandy regolith layer 0-2 m thick which is made up of sandy-clay of quartz grains, 2) A laminated saprolite layer of variable thickness of 10 - 15 m derived from in-situ weathering of granite, 3) A 15-20 m thick layer of fissured granite, where weathered granite and some clay partially fill the fissures. The effective porosity of this layer is very low and mainly due to the fissure zones (Dewandel et al., 2006; 2012).

A topographic and magnetic survey was conducted in the region. The geo-electric measurements at CPL were based on orthogonal 500 m electric dipoles and magnetic pulsations were measured with solid core induction coils. Hourly values of the



magnetic variation and analysis of equatorial magnetic pulsation were reported from
CPL (CSIR NGRI report, 1972). These hourly values are published in the Indian
magnetic data volumes and uploaded to WDC Kyoto (Svendsen et al., 1990). Figure
1b shows the 105 acre star shaped campus of CSIR-NGRI in Choutuppal. One high
magnetic anomaly is present at the eastern part of the campus. In the rest of the area,
the total range is about 80 nT. The surface topography is least in the east and north
and highest is the west and southern part of the campus. Several shallow boreholes
drilled in the northern end are used for hydrogeological studies in fractured hard rock
terrains. These studies monitored the nature of the granitic basement rocks, local
hydrogeology, and groundwater recharge within the CPL observatory.

Consequent to the Metro Rail project in the vicinity of the HYB magnetic observatory
in Hyderabad, the Choutuppal campus was re-visited for re-location possibilities of
HYB. This work showcases the different situations, which affect the operation of a low
latitude magnetic observatory, some mitigation measures and some unanswered
questions.



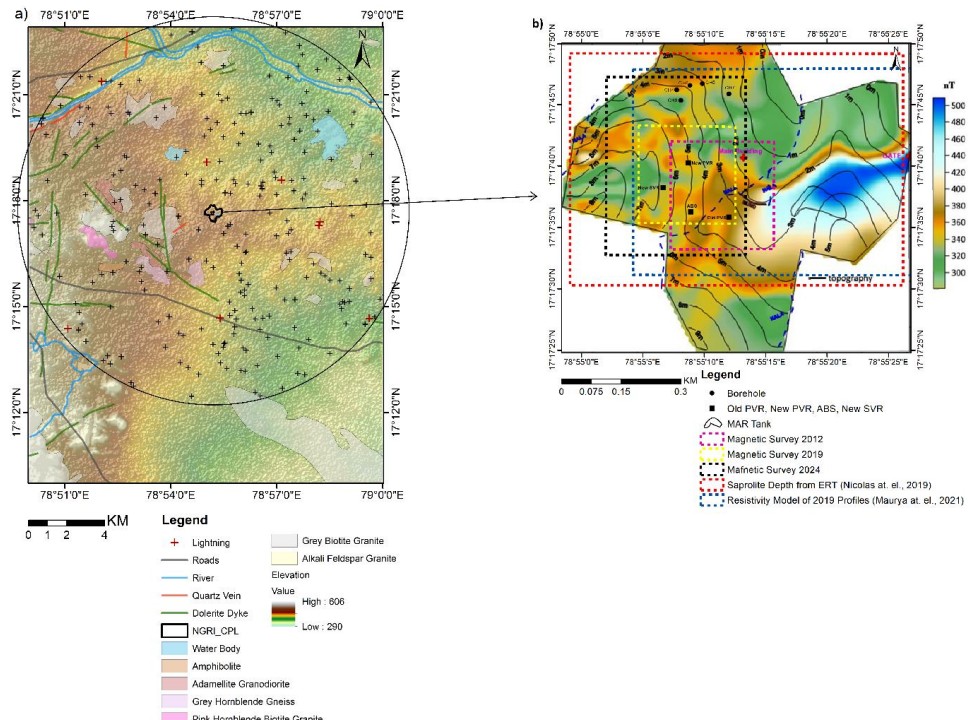


**Fig. 1:** a) The geological map shows the area around the NGRI- CPL observatory superimposed over topography (SRTM Global 30 meter) situated on alkali feldspar granite, plus symbols show lightning locations within a 10 km radius (circle mark) from January 2022 to August 2022 with maximum intensity 60480 amp, b) The magnetic anomaly map (after Shankar Narayan et al., 1967) superimposed over the elevation with marked area of previous studies. Magnetic anomaly plot of the sub region for year c) 2012, d) and 2019 respectively. Black box is the marked location for the New PVR in figure 1d. PVR=Primary variometer room, NV=New Vault, ABS= Absolute room, SVR= Secondary variometer room,

## 2. First phase of CPL Magnetic Observatory:

### i) Survey of magnetic gradients and building CPL Observatory

Prior to establishing the observatory buildings, a magnetic survey was conducted in

November 2012 over an area of 200 m x 200 m with 20 m intervals, between the Main

building and the southern side of the campus, marked by pink dotted line in Figure 1b,

which appeared to be have smallest anomalies from the earlier survey. This area was

sufficiently far away from the boundary of the campus to ensure that local activities

outside the campus may not have significant contribution to the measurements. The
total magnetic anomaly range (Figure 2) was ~ 150nT with changes in magnetic field
within ~20 nT around proposed locations of the PVR and ABS rooms.

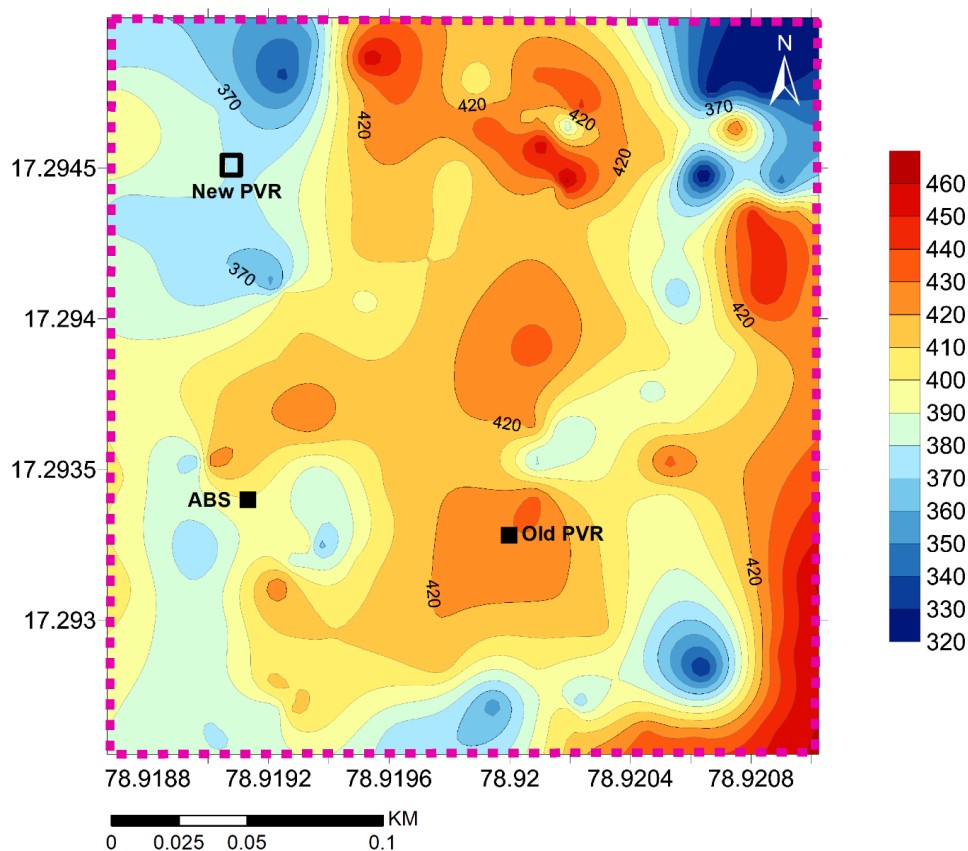


**Fig. 2:** Magnetic anomaly plot of the survey region in 2012. Black box is the marked
location for the New PVR.


In this central location the surface topography was moderately low and a shallow water
channel (nalla) flowed SE-NW through the area between the PVR and ABS. The old
PVR construction (with double walled, underground vault) started in the June 2013
using non-magnetic sandstone; the ABS room was constructed on slightly elevated
ground  with  two  pillars  inside  and  four  pillars  outside.   In April 2014 the  CPL



observatory was commissioned, equipped with tri-axial fluxgate magnetometer and
Zeiss single axis fluxgate theodolite for Declination-Inclination measurements. The
XVII IAGA Observatory Workshop was held on this premises in October 2014 with 93
international participants from 33 countries. The definitive data from CPL is published
at INTERMAGNET 2015 onwards.

**ii)     Hydrogeological Park and managed aquifer recharge:**
The region of CPL Observatory has a semi-arid climate with an average annual
maximum temperatures ranging as 28°C - 45°C. The mean annual rainfall is around
751 mm, which ranges from 2 mm in February to 171 mm in July. Water levels are
highly variable depending on the monsoon and usually range between 2 and 26 m (m
bgs). Water level measurements at the northern part of the Choutuppal campus has
been monitored since 1999 in two dozen boreholes, shown in Figure 1b,  by the Indo-
French Center for Groundwater Research (IFCGR) (Mareschal et al, 2018) to study
the hydrodynamic properties and associated hydrological processes in crystalline
aquifers. As part of a governmental scheme of strengthening groundwater through
recharge state-wise MAR projects, an infiltration basin was dug in Choutuppal (marked
in black outline as MAR in Figure 1b) during 2015 to meet the demands of farmers in
the area facing water scarcity. The basin has dimensions of 120 m by 40 m, with a
depth of about 2 m, effectively removing the regolith layer and extending into the
saprolite. The basin is mainly supplied by a canal which deviates water from the Musi
River. (Nicolas et al., 2019; Maurya et al., 2021) and was first filled in 2015. Nicolas et
al (2019) has shown that intra-seasonal groundwater fluctuations have only moderate
response to rainfall patterns, which could be due to usage trends as well as hydraulic
permittivity parameters. After the MAR basin filling, groundwater levels rose by 6 m in





one year. Figure 3a shows the water level changes of five boreholes before and after
monsoon from 2011 to 2023 with a data gap during 2021-22. While it is clear that most
of the time water level lies at intermediate depth of ~ 10 to 30 m, individually, CH5 and
CH7 show the least seasonal fluctuations over the years, CH6, CH4 and CH9 show
variations of 20 m or more; possibly very local fracture properties facilitate flow of
downward seepage of water preferentially. Starting from April 2017, the water levels
in the boreholes rose significantly, coming almost to surface in September 2017 and
reducing a little by July 2018. In December 2023, the water levels recorded are nearly
similar to that of September 2017. It can be inferred that sustained water in the MAR
basin has allowed the shallow aquifers to be permanently recharged. In September
2017, the rainfall of the monsoon combined with prevalent saturated condition led to
the flooding of the magnetometer room. The vault of the PVR rose nearly to surface
and damaged the fluxgate magnetometer and electronics. The water level receded by
a few metre the following summer but did not fall to earlier levels. While this was good
news for recharge, the PVR was made unusable.





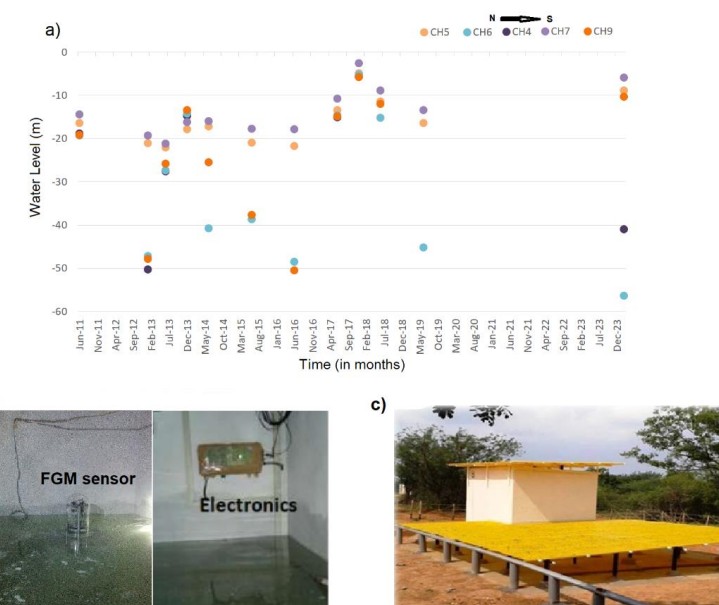


**Fig. 3a:** shows fluctuations of water level in a few boreholes in the north of Choutuppal campus. **Fig. 3b&c** show the outside view of the PVR and the submerged vault in September 2017.


### 3. Second phase of CPL Magnetic Observatory:

**i)     Survey and commissioning of new PVR**
With the need for a new variometer building, a fresh survey was conducted in May
2019 using GSM19 Overhauser magnetometers, marked by yellow dashed box in
Figure 1b. This area was some tens of meter west of the earlier survey location,
still central to the campus, on ground which is about 2-3 m higher. This total field
magnetic survey was carried out during six days (geomagnetic quiet days) in May
2019 over a 240X240 square meter area with 10 m spacing. The magnetic anomaly
of the region shows total amplitude variation of ~300nT (Figure 4a) with ~10nT
anomaly north of the ABS room; the proposed location is marked as New PVR in





Figure 4a. This time a raised building with double walls and double roof was
constructed of non-magnetic limestone to ensure no groundwater incursion issues
for the foreseeable future (Figure 4b). The New PVR was commissioned in January

167    2021.

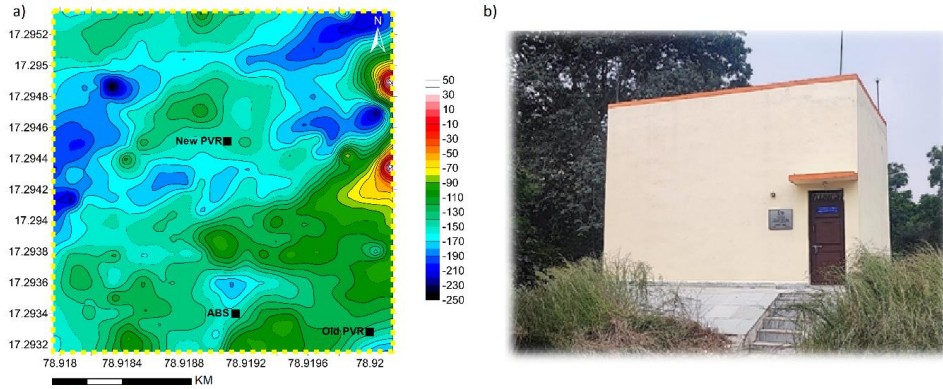



**Fig. 4:** Magnetic anomaly plot of the survey region in 2019. New PVR building
inaugurated in January 2021.


Recurrent malfunctions of the electronics (2-3 times a month) of the digital fluxgate
magnetometer (DFM) recording system were noticed in second half of 2021 and 2022,
a new phenomenon. These incidences were later correlated with lightning activity in
the vicinity of the Choutuppal campus. Being an open ground with no tall obstructions,
the lightning activity in this area was found to be more frequent than around HYB
observatory in Hyderabad. Strengthening of the earthing pits and lightning arrester
system only marginally countered the effects on the electronics.

**ii)    Lightning activity patterns around CPL Observatory and effects on**
**data**





ISRO-National Remote Sensing Center (NRSC) has a network of 27 VLF lightning
detection sensors, covering the southern part of the country (shown in Figure 5a).  The
detection range is upto 800 km with more than 98% confidence within 300 km range,
with 50% overlap to maintain high geo-location accuracy. on the  recent past which
may be due to the induced current generated in the subsurface. We have examined
the lightning data in a radius of 10 km around CPL observatory, marked as + in Figure
1a. Figure 5b shows the shows the occurrence frequency of lightning over the months
from January to August 2022; + symbols denote the instances when the lightning
caused damage in the fluxgate electronics. Figure 5b shows that substantia lower
intensity lightning activities are recorded during January, April, May and June.
Surprisingly July had no lightning in the area in 2022, in August, the higher intensity
lightnings were more numerous. Two instances of failure of the instrument electronics
occurred during the higher intensity lightnings of April and August, whereas a
disturbance of May was associated with lower amplitude lightning, discussed in next
section.

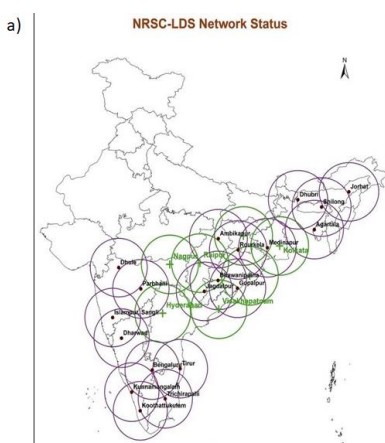 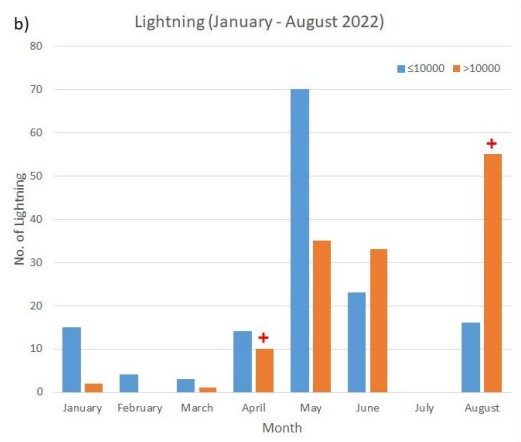





**Fig 5: a)** Map of lightning sensors **b)** Distribution of low and high intensity lightning
occurrences during January to August 2022.

Examination of the H, D, Z components of 1-sec data in LT for 04th May, 2022 data
from the new PVR shows continuous spikes from 07 LT to 8.2 LT in all the components,
followed by a shift of ~ 70, 20 and 150 nT in H, D and Z respectively (Figure 6a). After
rebooting the instrument, the data came back to its normal range. Comparison with
lightning data established that this disturbance was due to lightning effect (correlated
red mark).


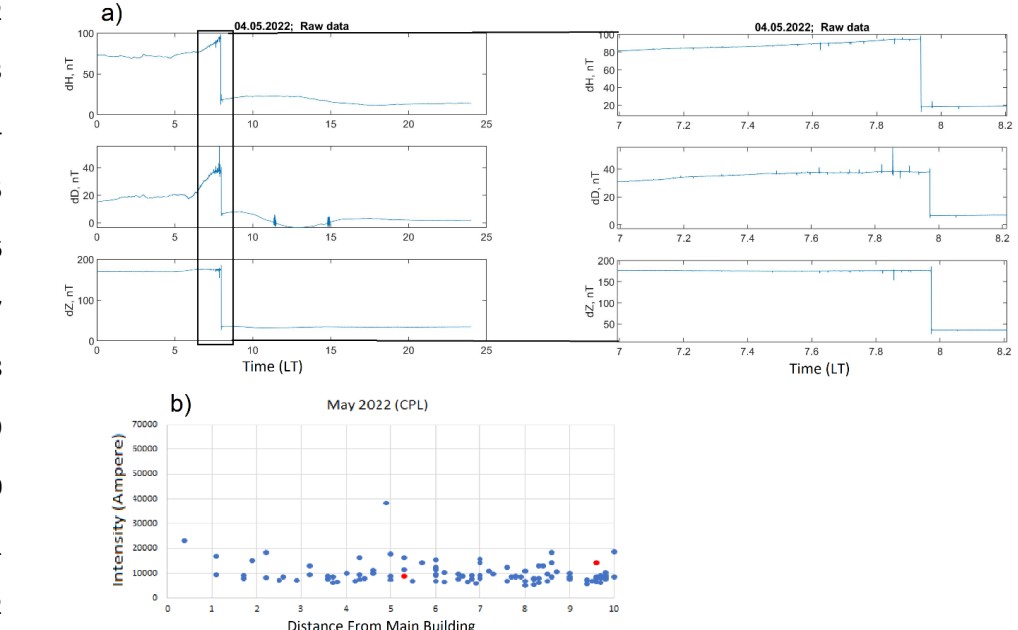

**Fig. 6:** a) Raw data (H, D, Z component) plot for 04th May, 2022, b) plot of lightning
intensity with distance from main building.

Further, we examined the H, D, Z components of 1-sec data fin LT or the 30th April and
26th May, 2017 data from the old PVR, on days which had some weather activities.



From the data, it is clear that there were no shifts in the data, but some continuous
spikes were observed from 18 LT to 18.8 LT (Figure 7). The spikes are more prominent
in the Z component (>0.5nT). Though lighting data was not available in this duration,
the general conditions lead us to believe that these minor signatures were lightning
induced.


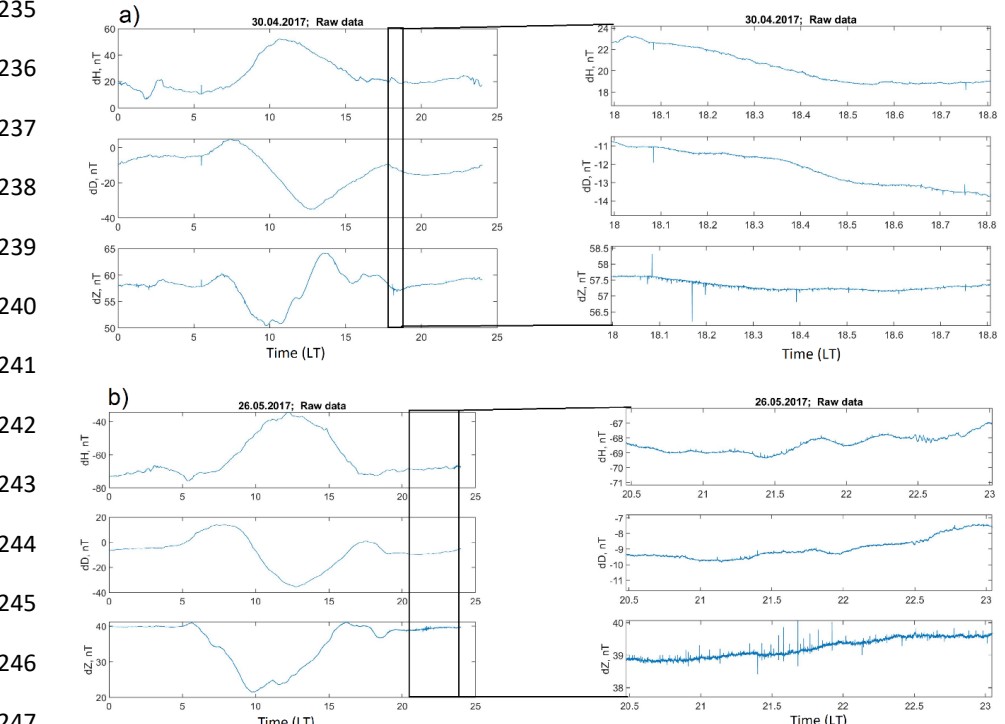

**Fig. 7:** a) Raw data (H, D, Z component) plot for a) 30[th] April, 2017, b) 26[th] May, 2017
at the old PVR.



Table 1 provides examples of the amplitudes of the disturbances recorded in data vis-
à-vis the light intensities and distance from the recording system



| Date | Time (LT) | H | D | Z | Lightning amplitude (ampere) | Distance (km) | Remark |
|---|---|---|---|---|---|---|---|
| 15-04-2022 | 17.7 – 18.8 | 0.1nT | 0.2nT | 1nT | 20968 | 5.5 | GSM90 stopped recording & spikes in HDZ |
| 04-05-2022 | 1.4 – 2.5 | 70nT | 20nT | 150nT | 8879, 14243 | 5.3, 10 | Shift in data(HDZ) |
| 29-08-2022 | | | | | 37387,24329, 21210, 8553 | 3.9, 5.5, 9.2, 10 | DFM Stopped recording |
| 30-04-2017 | 12.2 – 13.4 | 0.8nT | 1nT | 1.2nT | | | Spikes in HDZ |
| 26-05-2017 | 15.0 – 17.5 | 0.2nT | 0.2nT | 1nT | | | Spikes in HDZ |


It can inferred that the location of the new PVR as well as the fact that the pillars and
infrastructure were installed in the surface layer, instead of 3-4 m deeper, has amplified
the effects of lightning activity on the data. Given the fact that in future years more
uncertainty and swings in climatic parameters are anticipated due to effects of climate
change, it was deemed necessary to conduct further studies to find a location based
on optimal ranges of a variety of parameters like topography, distance from MAR lake,
the configuration of near surface regolith and saprolite layers along with groundwater
conditions.

**4. New search for optimal location**

Higher ground away from the MAR basin can be found toward the western side of the
campus. This study aims to delineate the sub-surface structures using the high-
resolution magnetic data in combination with resistivity information conducted through
Electrical Resistivity Tomography (ERT) and Electrical Vertor Resistivity Imaging
(EVRI) measurements at the Choutuppal campus and surrounding areas during 2016-
2017. The magnetic method can be sensitive to presence of near surface variations



and produce a model of these layers as well as in locating faults, folds, shear zones,
delineating geological structures, and groundwater contamination studies (Reynolds,
1997; Hinze et al., 2013; Kumar et al., 2018; Dwivedi and Chamoli, 2021,
2022). Finally, we try to find out a suitable location called new SVR (secondary
variometer room) where effects of lightning and groundwater level changes can be
minimum in the medium-long term.

**i)    2024 survey**

A 1-sec total magnetic field survey was conducted during February 2024 (4 days) at
10 m sampling intervals as same done in 2019 survey. The data of 2019 and 2024
were combined, thus covering a total of ~ 400x320 square meters. The diurnal and
International Geomagnetic Reference Field (IGRF 14) corrections are applied to the F
data, so that the residuals reflect only the local crustal contributions. Finally, we applied
the kriging interpolation method to generate the grid and produce the magnetic
anomaly of this area (Figure 8a). Further, this was converted by a 'reduced to equator'
computation to remove ambiguities in location of causative sources of magnetic
anomalies, at low and high latitudes. In this study, we chose the values of
Delineation=0°, and Inclination=24° and estimate the RTE generated magnetic
anomaly map of the study region. The RTE filtered map centres anomalies over their
sources and removes the asymmetry of the magnetic anomaly due to nonzero
magnetic inclination and helps in magnetic data interpretation (Figure 9).



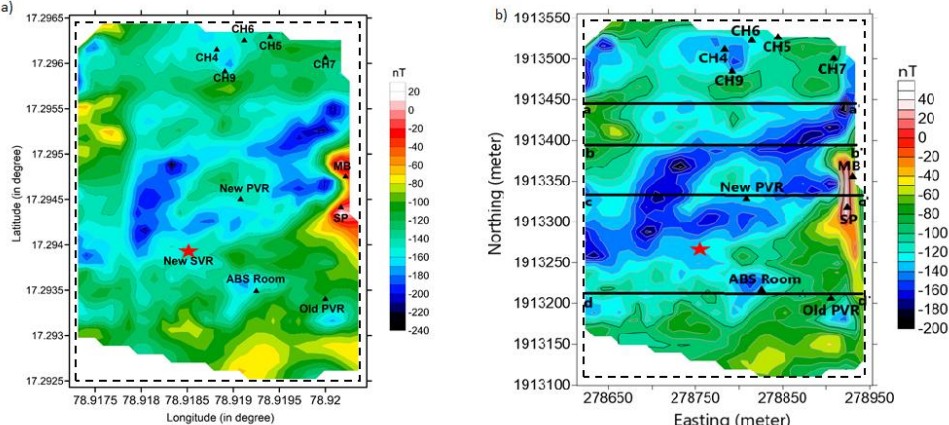

**Fig. 8: a)** The magnetic anomaly of the study area. (MB=main building, SP= solar panel, CH= bore well channels, ABS= absolute, PVR= primary variometer room, SVR= secondary variometer room). Red star mark shows the proposed New SVR in the region. b) the magnetic anomaly after reduction to equator of the study area. The aa', bb', cc' and dd' shows four magnetic profiles modelled.

The RTE magnetic anomaly shows a variation of ~ 260 nT in the region. The low anomaly is dominant in the central part followed by the high anomaly near the main building and solar panel. The new PVR lies in the low anomaly region where the three-axis flux magnetometer is installed to record 1-sec H, D, and Z component data of Earth's magnetic field. The ABS room is set up in the low anomaly zone to measure the Delineation-Inclination using the Mag-01 DI-fluxgate magnetometer.

We have considered four profiles aa', bb, cc', and dd' along EW in the RTE magnetic anomaly map to characterise the subsurface susceptibility model (Fig. 6). The magnetic data shows its importance in characterizing the shallow sub-surface structures, which would be further beneficial for the selection of new location to install magnetic observatory in the campus. The lightning data is considered from the National Remote Sensing Centre (NRSC), Hyderabad, India, to investigate the failure of the DFM electronics during the lightning strike. The high-resolution topography data





318 is obtained from the Shuttle Radar Topography Mission (SRTM) Global 30

319 (https://earthexplorer.usgs.gov/) to plot the elevation of the region.

320

321 **ii)  Analysis of subsurface source and depths**

322

323 The fast Fourier transform (FFT), a robust technique is used by several researchers

324 to calculate the mean depth of layered interfaces of the potential field datasets

325 (Chamoli et al., 2011, 2023; Dwivedi et al., 2019). The power spectrum analysis gives

326 the average depth of the sources with an error limit of 10% (Mishra and Pederson,

327 1982). The 2D radially averaged power spectrum of the RTE magnetic anomaly data

328 shows two linear slope segments corresponding to the average depth of two interfaces

329 around 12±1.2 m and 1±0.1 m (Figure 9).

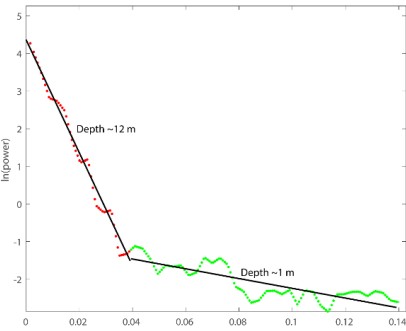

331 **Figure 9:** Radially averaged power spectrum of the RTE magnetic anomaly. Different
332 layer segment give an average depth of layered interfaces.

334 The tilt depth is an effective method in characterizing the location of source edge as

335 well as magnetic source depth using the tilt angle (TDR) approach (Salem et al., 2007).

336 First, the TDR is described by the following equation (Miller and Singh, 1994).



$$\theta = tan^{-1}\left[\frac{\partial M}{\partial Z} \middle/ \sqrt{\left(\frac{\partial M}{\partial x}\right)^2 + \left(\frac{\partial M}{\partial y}\right)^2}\right]$$ (1)
where $\frac{\partial M}{\partial x}, \frac{\partial M}{\partial y}, \frac{\partial M}{\partial z}$ are the first derivative of the magnetic field M in the x, y and z
directions. The zero contour ($\theta = 0^0$) demarcates the spatial location of magnetic
source and tilt amplitudes restrict in the range of $+90^0$ to $-90^0$. Salem et al., (2007)
introduced the tilt depth technique using the relationship among tilt angle ($\theta$), depth
($Z_c$), horizontal location ($h$) of a contact as:
$$\theta = tan^{-1}\left[\frac{h}{Z_c}\right]$$ (2)
In the equation 2, the contact location ($h = 0$) lies to the zero values of the contour and
depth relates to the horizontal distance between the $0^0$ and $\pm 45^0$ contour in the TDR
map. We apply the technique to generate the tilt angle (TDR), tilt depth solution and
histogram plot on the magnetic gridded data. Figure 10 presents the TDR map with
displaying of contours of $-45^o$, $0^o$, $+45^o$ (dashed black lines). The TDR shows the short
wavelength anomalies and closely spaced contours that corresponds to shallow sub
surface sources. The zero contour values of the TDR show the location of the source
whereas the half of the distance between $\pm 45^o$ contours demarcates depth of the
sources. It can be seen that distance between $0^o$ and $+45^0$ is demarcated by relative
closeness over the shallow sources and wide expanses over the deeper sources.










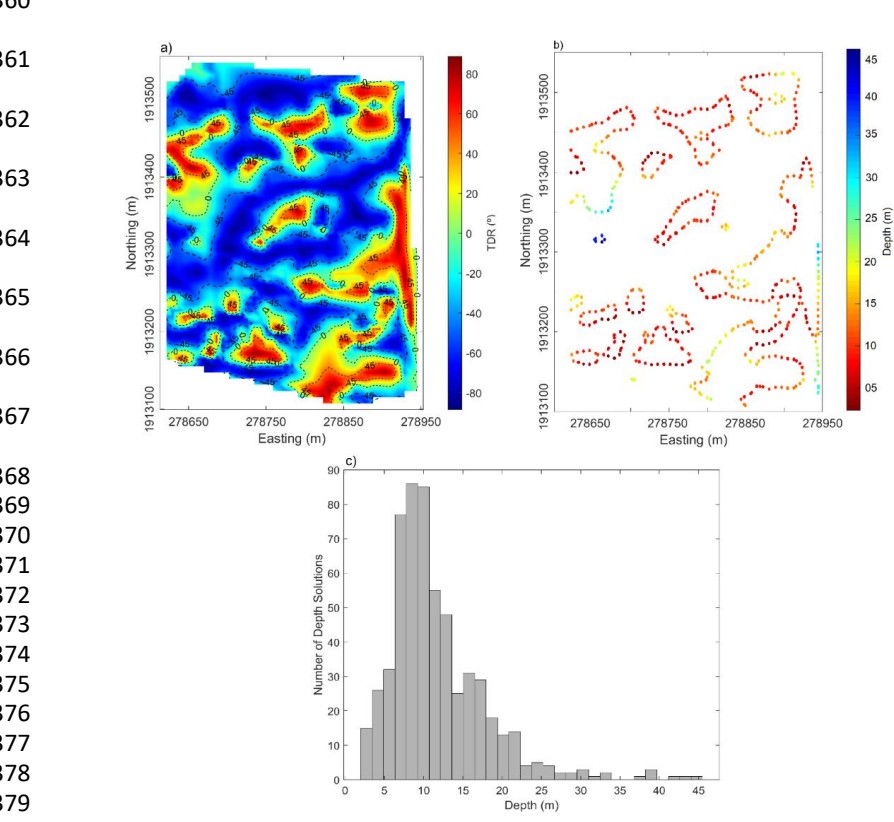

**Fig. 10:** a) The tilt angle plot, b) The tilt depth solutions, c) the tilt depth solution histogram plot of the RTE magnetic anomaly.

The tilt depth solutions of the RTE magnetic anomaly data with depth variation from ~ 2 m to 45 m. Most of the sources lie in shallow depth ~ 2 to 20 m and extended in one direction. The histogram plot shows the variation between number of depth solution and depth (Fig. 8c). It is clear from the plot that number of depth solution are maximum in the depth range of ~ 2 m to 20 m which corresponds to shallow source in the sub surface.

**iii)    Magnetic data inversion and forward models**



We invert the RTE magnetic anomaly to estimate the depth variation of the interface
with strong susceptibility contrast.  We use the method of Parker (1972) which works
in the Fourier domain to estimate the depth variation of an undulated interface. The
depth of interface can be computed from the magnetic anomaly due to an uneven,
uniform magnetized layer by inversion procedure. The method is improved by
incorporating high cut filter to ensure the convergence of series and to avoid instability
at high wavenumber (Pham et al., 2020). Based on the power spectrum
characteristics, we have chosen the cut-off wavenumber between 0.038-0.13 m$^{-1}$ to
remove the high frequencies. The resultant map shows that this interface is deepest
in the centre of the survey area and shallowest towards the edges; then present PVR
and the ABS Room are in the zones where this interface is deep, whereas the old
PVR, which was flooded was in the zone of shallow interface  (Figure 11a). The
calculated depth is inferred as the saprolite layer, which varies from ~ 12 to 16 m with
a mean depth of ~14 m. Figure 11b shows the variation of the root mean square (RMS)
error against the iteration number. In this case, the inversion process performed 175
iterations, and the RMS error between two successive approximations was reduced
from 0.0680 m to 9.9584× 10$^{-5}$ m.

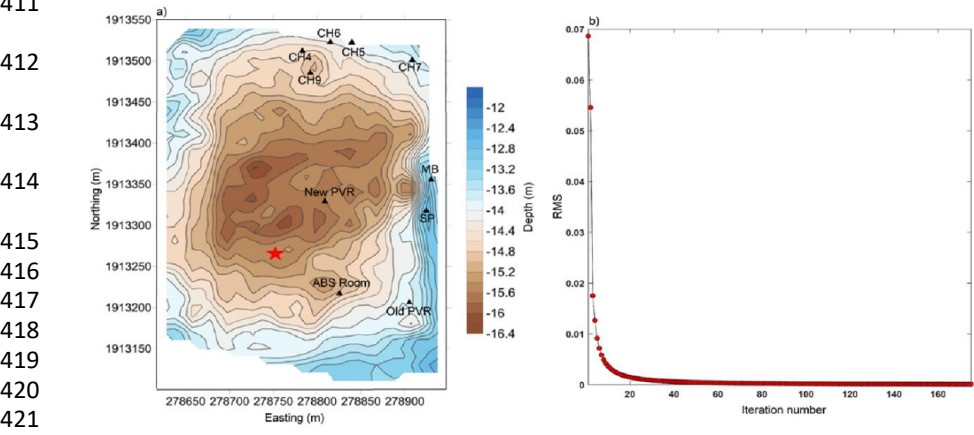



**Fig. 11:** a) The depth variation of the saprolite interface derived from the inversion of
the RTE magnetic anomaly after removing the high frequency component, b) variation
of RMS error against the iteration number.
Further, we model the RTE derived magnetic anomaly data along four profiles aa', bb',
cc', and dd' (Figure 8b) to delineate the details of the sub-surface structures. The
forward modelling along these profiles is carried out using the IGMAS+ software, a
tool for forward and inverse modelling of potential field datasets (Anikiev et al., 2023).
The total length of theses profiles are ~310 m which show the depth variations up to
50 m from the surface (Fig. 12a, b, c, d). The 2D models along these profiles explains
the presence of three layered structures from the surface up to a depth of 50 m as:
sandy regolith (~0.3 susceptibility in SI), saprolite (~3 susceptibility in SI), and fissured
granite (~2.5 susceptibility in SI). The average susceptibility value for sandy regolith
layer is measured in the field using KT-10 magnetic susceptibility meter whereas
others are referenced from Telford et al., (1990). The saprolite interface is incorporated
in the models based on previous results of ERT data (Nicolas et al., 2019). The
average thickness of the sandy regolith layer is ~3 m whereas the saprolite layer is ~
10 m in the models. Both the saprolite and regolith layers show undulations in the all
models.











**Fig. 12:** Magnetic data modelling across the profiles i) aa', ii) bb', iii) cc' and iv) dd'
incorporating constrain from power spectrum, and past ERT studies. Top, middle and



bottom layers in the models are sandy regolith, saprolite and granitic basements. The
marked arrows show the location of New and Old PVR in profile cc' and dd'.
**iv)      Correlation with conductivity data/information**
The cc' magnetic profile is close to the vertical resistive cross-section along AA' of
Maurya et al., (2021) where resistivity increases from surface to depth (Fig. 13a). The
old surface and apparent resistivity data of the region is digitized from the Sanker
Narayan et al., (1967) and overlaid over the topography (Fig. 13b, 13c). The apparent
resistivity shows lateral varying high resistivity outside and vice-versa inside the
Choutuppal campus (Fig. 13c). This means that conductivity decreases away from the
campus laterally and reflects the presence of conductive soil inside the campus.

The calculated saprolite depth by inverting magnetic data at different locations are ~
15.5 m (new PVR), ~ 15 m (ABS Room and CH9), ~ 13 m (MB and SP), ~14 m (old
PVR, CH5, CH6, CH7), ~ 14.5 m (CH4) (Fig. 11a). Whereas, depth estimation by the
ERT data are ~ 13.5 m (new PVR), ~ 18 m (ABS Room), ~ 15 m (CH9), ~ 14 m (MB,
SP and CH4), ~22 m (old PVR and CH6), ~ 29 m (CH5) ~ 17 m (CH7) respectively
(Fig. 11d). The calculated depth from these two different datasets show a discrepancy
of ~ 2m except at the locations of CH5, CH6 and old PVR. The ERT survey estimates
the upper fissured layer depth (~ 9 to 33 meters) of the Choutuppal campus and shows
the undulated interface creating compartmentalised aquifers (Nicolas et al., 2019).





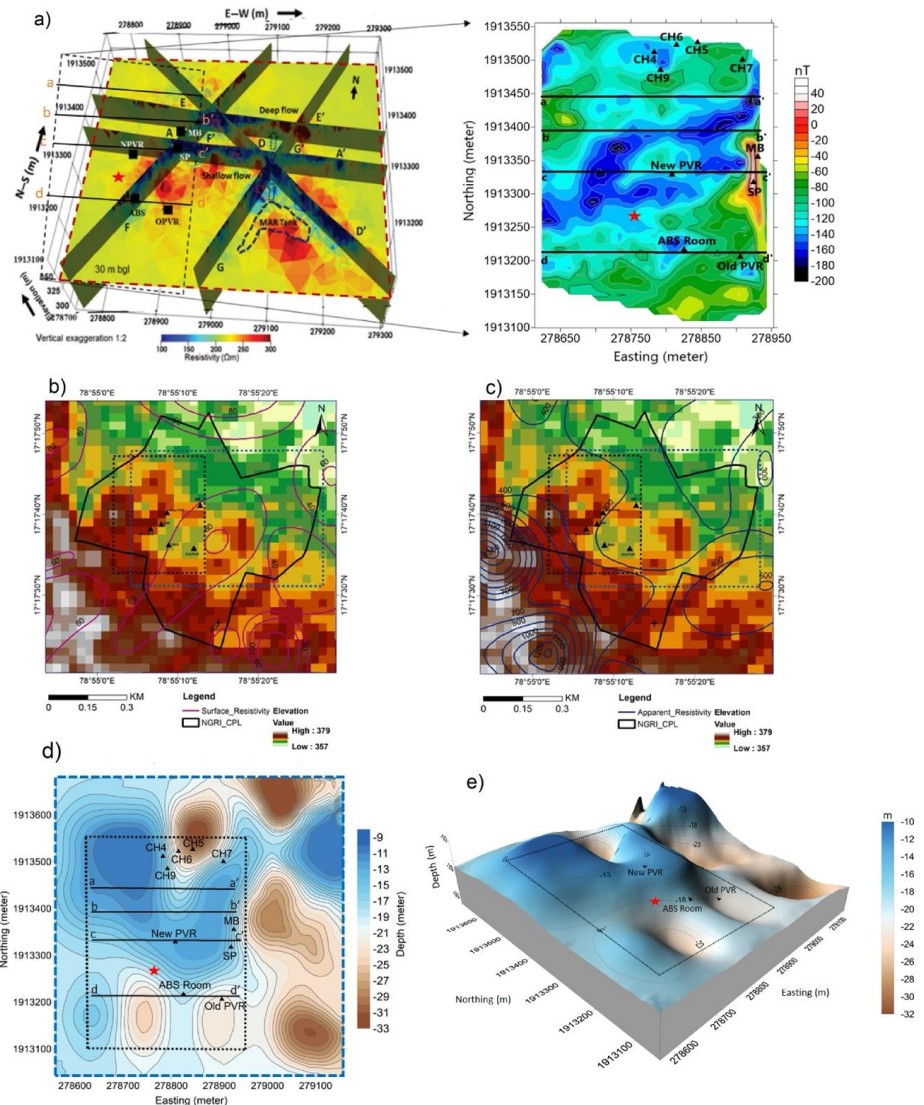

**Fig. 13:** a) A resistivity model of the 2019 profiles along AA', DD', EE', FF', and GG' and a depth slice at 30m below ground level (after Maurya et al., 2021), b) surface resistivity, c) apparent resistivity superimposed over the topography with lightning strike location (× symbol, intensity of 23208 amp. at a distance of 400 m from the MB on 04th May, 2022). The blue dotted lines is the area correspond to resistivity model of 2019, d) depth estimation of saprolite interface from ERT data (after Nicolas et al., 2019), e) 3D view of saprolite interface from figure 11d, e) a glimpse of newly constructed PVR building, f) installed DFM on non-magnetic pillar above the surface. Dashed black line shows the magnetic anomaly region



**5. Discussion and conclusion: proposed optimal location for SVR**
The high resolution magnetic data provide a detailed shallow sub-surface structure at
the CPL observatory. The power spectrum result shows two segments at a depth of ~
12 m and ~1 m corresponding to saprolite and sandy regolith interface. These depths
are similar to the previous depth estimated using the drilling data by Dewandel et al.,
(2006). The inversion results show the depth variation of the saprolite interface ~ 12
to 16 m (Fig. 12a) which underestimate with the upper fissured layer depth from the
ERT survey. The obtained interface shows depression in the central region of the map
with crests in the outer region and linear variation in depths (Fig. 12a). The estimated
depths at different locations shows ~ 2 m differences from these two datasets except
at the location of CH5, CH6 and old PVR. These discrepancies in the depths
estimation may be due to measurements of two different independent physical
parameters susceptibility and conductivity. The saprolite interface might be delineate
better by the ERT data due to presence of significant resistivity contrast between
saprolite and granitic bedrock in the region (Robinson et al., 2008; Gourdol et al.,
2021). The tilt depth plot of the anomaly data reflects depth variation from ~ 2 m to 45
m (Fig. 8b). The histogram plot confirms the presence of the shallow sources in the
depth range of ~ 2 m to 20 m based on the majority of a number of depth solutions.
These shallow sources are in circular and elongated shapes which may be residue of
igneous intrusion (Gorczyk and Vogt, 2018). At shallow depths, circular bodies with a
magnetic contact source might produce symmetric anomalies in the magnetic data
(Hinze et al., 2013). In contrast, elongated bodies may produce linear magnetic
anomalies that follow the direction of the body as in the present study.



The sub-surface susceptibility models along the magnetic profiles aa', bb', cc', and dd'
reflect the geometry of sandy regolith, saprolite layer and basement fissured granite
(Fig. 13). The saprolite layer shows undulating variation with quiet thick and thin at
different locations along the profiles. These variations illustrate that the saprolite layer
is thin in the region where the anomaly is low and it shows the thickness in the region
where the anomaly is high. The magnetic anomaly shows a large depression of length
~ 80 m in the profile cc' (Fig. 12c) where saprolite layer is absent and this depression
arises due to sandy layer in the model. The 3D cross-sectional resistivity model infers
the presence of high conductive anomalies followed by the moderate conducting
saprolite layer and the low conductive basement rocks (Fig. 12a).

Figures 12c and d show that the apparent resistivity increases laterally away from the
campus and shows the presence of high conductive soil layers inside the campus
(Sanker Narayan et al., 1967). The apparent and surface resistivity demonstrate the
linear relationship with the topography. The comparison between the latest resistivity
model (~200 Ωm variation) for 2019 profiles (Fig. 11a) and the old apparent resistivity
(~300 Ωm variation) plot in 1964 (Fig. 11c) shows the resistivity change of ~100 Ωm.
This discrepancy may be due to the presence of newly constructed artificial recharge
pond which may decrease the resistivity due to its high conductive nature. The thin
saprolite layer corresponds to high conductivity, which gives low anomaly of the
magnetic data and vice versa. Overall, the resistivity variation increases from surface
to greater depth. The resistivity model also show that the artificial recharge pond has
good subsurface connectivity, helping the groundwater recharge towards the north and
northwest directions in the campus (Maurya et al., 2021).



The water level started to rise in June-2016 and reached to very shallow depth in
channels during December 2017. It is evident that since last half of 2016, the recharge
has led to saturation, which transformed the hydrogeological regime of the campus.
The rainfall of 2017 monsoon combined with the already prevalent saturated
conditions led to the flooding of the magnetometer vault. The location of new PVR has
advantages as it is away from the water recharge pond, minimal magnetic gradient but
generation of induced current in the rainy seasons due to presence of conductive
environment around the location.

The effect of lightning strikes on the data with increasing distance, intensity and ground
conductivity shows that higher intensity strikes has had an impact of the data and
instruments. Based on the above it is very crucial to determine the location and
configuration where the installation can avoid the effects groundwater fluctuations as
well as lightning strikes, based on the nature of subsurface rocks, soil conditions, and
their magnetic variations. The susceptibility model along with conductivity information
are used to make a selection of a new site for trial measurements, indicated by red
star in Figures 8, 11, 12. This location is on moderately high ground, depth of saprolite
late is about 20 m, conductivity in range of 200 $\Omega$m, magnetic gradient of ~20 nT. It is
proposed to construct the pillar in a semi-underground vault below regolith level, to
avoid additional currents during the rainy season as well as during lightning strikes.
The surrounding area around the pillar should be completely resistive to minimize the
generation of these induced currents. The more thickness of saprolite may create
resistive environment in the region and water may also not create problem due to away
from the recharge tank due to sufficient distance.



**Acknowledgements:**
We thank the Director, NGRI, for supporting the work; reference no. …….. The authors
are thankful to Dr. Subhash Chandra and other colleagues from the ground water
department of CSIR-NGRI for providing the bore well water level data and resistivity
results. We also thank Dr. Phani Chandrasekhar for help towards the repeat surveys.

**Authors' contributions:**
**Divyanshu Dwivedi:** Conceptualization, Methodology, Computation and Modeling,
Formal analysis, Writing-original draft
**Sneh Yadav:** Data Processing, Modeling
**Kusumita Arora:** Conceptualization, Validation, Review and final editing
**Rakesh Murteli:** Lightning data analysis, maps and figures
**Alok Taori:** Lightning data, validation

**Declaration of Competing Interest:**
The authors declare that they have no competing financial interests or personal
relationships that could have appeared to influence the work reported in this paper.

**Funding:**
No funding was provided for this work.

**Availability of data and material**
The magnetic data associated with this research are available and can be obtained
upon the request from corresponding author. The topography data is obtained from



the Shuttle Radar Topography Mission (SRTM) Global 30
(https://earthexplorer.usgs.gov/).

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
