# Peer review of "Optimal site selection for Choutuppal geomagnetic observatory, based on geophysical evidences"

_EGUsphere, 2025_

## Author Response (AR3)

**RC1 Comments and response**

General:

There is a lot of information in this contribution. Unfortunately, the actual structure of the article makes it hard to follow the arguments and allows the reader to distinguish between what was done in the past and what was done recently.

I suppose some kind of timeline or table in the introduction to provide the reader with information on the chronological order of surveys/information/decisions.

The pictures should be some kind of high resolution or in a vector format which scales well.

**Reply: We thank the reviewer for critical reading of our manuscript which helped us to improve the revised manuscript. Our specific replies to the comment are in bold blue. We have attempted and explain the raised queries as well as incorporated them in the track change mode in the manuscript. We already added a few lines to clear the flow of the manuscript (lines number: 73-76) and kept the high-resolution image that will help to reader.**

Figures:

1: c) d) missing, figures are hard to read.

**Reply: There is no figure 1c and 1d. It was a typo mistake. Corrected in the mansucript.**

 2: no unit at colorscale

**Reply: Incorporated the unit (nT) at colorscale.**

 4: a) no unit at colorscale

**Reply: Incorporated the unit (nT)  at colorscale.**

6: b) distance unit missing

**Reply: Added the distance unit (in km).**

 8: should be same nT range for better comparison

**Reply: The Figure 8a is the magnetic anomaly of the data whereas the Figure 8b shows the RTE filtered map of the Figure 8a. Thus, both plot would show different range. But the total anomaly ranges are same which is ~260 nT for both the cases. We have not done any comparsion in both plots.**

11: b) better log-log plot

**Reply: Revised the figure as suggested.**

13: lots of information, pictures need to be bigger

**Reply: Revised the figure as suggested.**

Introduction:

It is not really clear why a new site should be selected. The authors mention that there is a "Metro Rail project", but why they need to choose a new site is not explained. What are the consequences and what changes do they expect that will disturb the geomagnetic measurements?

**Reply: Geo-electric measurements on the CPL campus were discontinued in 1982, but when the Metro Rail project in the vicinity of the HYB magnetic observatory in Hyderabad appeared to threaten its existence in 2010, the Choutuppal campus was re-visited in 2012 for re-location possibilities of HYB. In September 2017, the rainfall of the monsoon combined with prevalent saturated condition led to the flooding of the Old PVR vault. After this incident, the operation at the low-latitude magnetic observatory like CPL has been affected by multiple factors such as lightning influence, varying soil conditions, the effect of nearby MAR tank, ground water condition, and local conductivity changes. Although some mitigation strategies have been applied, certain issues remain unresolved. Thus, all these factors collectively contribute to identifying the optimal location for the observatory where the effect of these parameters would be minimum.**

First phase of CPL Magnetic Observatory:

Line 97: What is PVR and ABS?

**Reply: PVR and ABS refer to primary variometer and absolute room respectively. The details about these abbreviation are already explained in the caption of Figure 1.**

Hydrogeological Park and managed aquifer recharge:

I suppose that CH5 to CH9 are the names for the boreholes? Please clarify. Where are they exactly? Ah they are in picture 8 … please reference as they come earlier.

**Reply: Yes, CH5 to CH9 are few boreholes in the area of study. We have mentioned the boreholes (CH5, CH6, CH4, CH7, CH9) in line numbers 140-141 of the manuscript. We have also added the Figure 3b which shows the location of boreholes in the map.**

Lightning activity patterns around CPL Observatory and effects on data:

There are two major flaws in this chapter which need to be explained better:
   1. Why the authors think that a shift in the data offset are due to the lightning activity? Please explain why this is so.

**Reply: The Lightning detection sensor network monitors the cloud-to-ground lightning occurrences by virtue of emitted waves in the 5 Hz - 30 MHz range and geolocation is calculated using the time of arrival method as elaborated by Taori et al (2022; 2023). These pulses can be detected by magnetometers like fluxgate. Changes in the ionosphere's conductivity (due to lightning) can indirectly influence geomagnetic field variations (Luque et al., 2025). From the table, it is clear that the generation of high lightning intensity caused the damage/shift/stop in recording systems at the same time for the event 4th May, 2022 (in local time), which supports that these disturbances arose due to lightning activity only not from any other sources.**

2. Line 257 to 259: "It can inferred that the location of the new PVR as well as the fact that the pillars and infrastructure were installed in the surface layer, instead of 3-4 m deeper, has amplified the effects of lightning activity on the data." Why is this? Why should a deeper installation neglect a lightning effect? And why should a surface soil layer amplify the effects of lightning? Please explain.

**Reply: It was suspected that because the new PVR is constructed on the surface and the cables were laid in the surface layer, instead of the vault configuration as in the one which was flooded. In the surface installation, the conductive nature of the soil creates a path for the propagation of the lightning current into the layer, amplifying the effects of lightning and thus increasing the likelihood of damaging the instruments.**

**The deeper installation of the magnetometer provided better data quality and minimize the effect of lightning and conductivity to increase the life of the instrument. However, as groundwater levels rose due to MARS, unfortunately this configuration had to be abandoned.**

New search for optimal location

Please explain why you choose ERT surveys to make a decision on a specific site which migrates the problems with lightning.

**Reply: A new site installation requires careful geophysical and geological investigation especially if we have to minimize the issues related to lightning-induced noise. In this context, the ERT survey plays a crucial role in providing 2D/3D images of subsurface resistivity. Soils and rocks with low resistivity conduct electric current in the subsurface. When lightning strikes the ground, it induces a transient electric current. These ground currents follow the resistive gradient. By placing the magnetometer on a resistive geological formation, we can minimize the coupling of lightning-induced EM fields.**

What is a RTE filtered map? Is it reduced to equator?

**Reply: Yes, a filtered map is reduced to an equator map. RTE is a data transformation technique applied to magnetic anomaly maps to simulate how the anomalies would appear if the survey area were located at the magnetic equator (where the inclination would be zero). At the low latitude region, it is very necessary to generate the RTE map to remove the asymmetric property of the anomaly from their actual source location that would be generated due to inclination.**

2024 survey

This chapter would benefit from a more detailed description of the ERT survey and its result, as well as a better picture of the conductivity result.

**Reply: We have taken the constraint from the ERT survey to gain better information about the resistivity distribution within the Choutuppal campus, which has already been published in the past (Nocolas et al., 2019; Maurya et al., 2021). In addition, we have written a brief description of the ERT survey in the revised manuscript (line number: 348-354). We have incorporated the high-resolution image of the resistivity variation (Figure 13a).**

Discussion and conclusion: proposed optimal location for SVR

In this chapter the authors explained the measurements in detail, but a discussion on the proposed optimal location for the SVR is almost missing. The chapter would benefit enormously from a discussion why these measurements lead to a decision.

**Reply: In the last pargraph the discussion and conclusion, we have discussed the proposed location for the new SVR. The susceptibility model along with resistivity information are used to make a selection of a new SVR (78.9185E, 17.2939N), indicated by red star in Figures 1, 8, 11, 13. This location is on low magnetic anomaly of ~ -145 nT (Figure 8a), resistivity ~200 $\Omega$m (Figure 13a), moderately high ground ~367 m (Figure 13b), and depth of saprolite layer ~ 20 m (Figure 13d). A thicker saprolite layer can enhance the resistive environment and reduce current propagation. The location's sufficient distance (~ 320 m) from the recharge tank ensures that water infiltration is unlikely to pose a significant issue. Based on these parameters, it is proposed that the pillar will be constructed within a semi-underground vault, in order to minimize the influence of induced currents during rainy seasons and lightning strikes. Additionally, the volume surrounding the pillar should be filled using high-resistivity material, such as Quartzite, to further minimize the likelihood of induced currents during lightning events or wet conditions (line number: 664-679).**

References:

1. Luque, A., Li, D., Bjørge-Engeland, I.,Lehtinen, N. G., Marisaldi, M., &Østgaard, N. (2025). Cumulative effects of lightning electromagnetic pulses on the lower ionosphere. Journal of GeophysicalResearch: Atmospheres, 130, e2024JD042121. https://doi.org/10.1029/2024JD04212

2. Nicolas M., Bour O., Selles A., et al., 2019. Managed Aquifer Recharge in fractured crystalline rock aquifers: Impact of horizontal preferential flow on recharge dynamics. Journal of Hydrology, 573, 717-732, https://doi.org/10.1016/j.jhydrol.2019.04.003.

3. Maurya V.P., Chandra S., Sonkamble S., et al., 2021. Electrically inferred subsurface fractures in the crystalline hard rocks of an Experimental Hydrogeological Park, Southern India. Geophysics, 86(5), WB9-WB18, https://doi.org/10.1190/geo2020-0327.1

4. Taori A., Suryavanshi A., Pawar S., et al., 2022. Establishment of lightning detection sensors network in India: generation of essential climate variable and characterization of cloud-to-ground lightning occurrences. Natural Hazards, 111, 19-32, https://doi.org/10.1007/s11069-021-05042-8

5. Taori A., Suryavanshi A., Bothale R.V., 2023. Cloud-to-ground lightning occurrences over India: seasonal and diurnal characteristics deduced with ground-based lightning detection sensor network (LDSN). Natural Hazards, 116, 4037-4049. https://doi.org/10.1007/s11069-023-05839-9

**RC2 Comments and response**

General comments:

The manuscript offers **comprehensive and valuable geophysical insights** for selecting an observatory site—it integrates magnetic surveys, ERT/EVRI resistivity data, hydrogeology, and lightning incident analysis effectively. The scientific scope and objectives are well-conceived and relevant to geomagnetic observatory planning. The manuscript may be particularly useful when designing observatories where the optimal measurement location must be selected in the presence of numerous risks.

After making some corrections, the manuscript can be published.

**We thank the reviewer for critical reading of our manuscript and for giving valuable suggestions. Our specific replies to the comment are in bold blue. We have attempted and explain the raised queries as well as incorporated them in the track change mode in the manuscript.**

**Specific comments:**

**Line 77:** MAR tank is marked, but it is not clear what object it represents.

**Reply: MAR tank is managed aquifer recharge. In hydrology, this is a surface water storage structure designed to collect and store rainwater or runoff. We have added in the caption of Figure 1.**

**Line 99:** The units of the X and Y axes are not specified. (More notes below.)

**Reply: Units of X (Longitude) and Y (Latitude) axes are in degree. Changed in the Figure 2 accordingly.**

**Line 117:** The meaning of (mbgs) is not clear to me.

**Reply: mbgs refers to "meter below ground surface". For example, 20mbgs= 20 meters below the ground surface. Incorporated in the line 125.**

**Line 120:** The meaning of MAR is unclear.

**Reply: Added in the caption of Figure 1.**

**Line 224:** What unit of measurement is the distance in Figure 6. b)?

**Reply: The distance is measured in kilometers. Added in the Figure 6b.**

**Line 298:** The anomalies in Figure 8. a) only partially correspond to the image in Figure 2. The scale and coloring are also different. To some extent, the comment is also true for Figure 4, but it is much better in line with Figure 8. It would be worth transforming Figure 2 or considering its possible omission. It is also possible that there was a partial change in the rock magnetism due to soil moisture ( https://www.annalsofgeophysics.eu/index.php/annals/article/view/7351). Note: The

carefully reduced values of the detected anomalies in Figure 8. a) are mostly negative values. This is most likely to be the case if one of the rocks has a remanent magnetism in the opposite direction.

**Reply: We have revised Figure 2 by updating the color scheme and adjusting the scale for improved clarity and visual consistency. The total magnetic anomaly range is ~150 nT (both positive and negative anomaly) which is the same as the previous Figure 2. The total anomaly range for Figure 8 is ~260 nT, which may be due to the large area in comparison to Figure 2. We have also examined the anomaly pattern trends in the overlapping region of both figures and found them to be consistent. It is evident that since the last half of 2016, the recharge has led to saturation, which transformed the hydrogeological regime of the campus. This may correspond to the partial change in rock magnetism due to water saturation (Csontos et al., 2019) resulting in a decrease in magnetic anomaly (more negative). We have added this useful information in the discussion section.**

**Line 308:** It would be worth emphasizing that the sentence is about the location and equipment for absolute geomagnetic measurements.

**Reply: We have revised the sentence as suggested.**

**Line 522:** "c) apparent resistivity superimposed over the topography with lightning strike location (✗ symbol, intensity of 23208 amp. at a distance of 400 m from the MB on 04th May, 2022)" I can not find the x symbol.

**Reply: We have used the + symbol throughout the manuscript. Changed the x by + symbol in the caption.**

**Line 526:** " e) 3D view of saprolite interface from figure 11d" Figure 11d does not exist. "e) a glimpse of newly constructed PVR building, f) installed DFM on non-magnetic pillar above the surface. Dashed black line shows the magnetic anomaly region" These sentences are not relevant here.

**Reply: Changed the caption according to Figure 13. Removed all the sentences which are not relevant here.**

**Line 599:** "magnetic gradient of ~20 nT" Does this indicate a difference between some pillars or that the total field change in some direction is 20 nT/m?

**Reply: The magnetic gradient of ~20 nT refers to the difference in magnetic values between the new PVR and the proposed new SVR (red star marked).**

**Technical corrections:**

**Line 44:** The accent in the reference is incorrect.

**Reply: Corrected the accent in the line 44.**

**Line 48**: Sankar → Sanker

**Reply: Corrected the spelling in line 48.**

**Line 120**: Mareschal et al, 2018 is missing from the references.

**Reply: Added the Maréchal et al, 2018 in the reference list.**

**Line 184:** The VLF abbreviation is not explained.

**Reply: Revised the sentence in line 200-201.**

**Line 186:** A link would be good for the listed technical parameters.

**Reply: We have cited the previous studies by Taori et al. (2022; 2023) in line 209, which provide detailed information regarding these parameters.**

**Line 205:** The marking of the H, D and Z components should be detailed. LT (Local Time) also.

**Reply: Adeed the detailed marking of the H, D, Z, and LT in line 227-228.**

**Line 312**: "subsurface susceptibility model (Fig. 6)" Incorrect figure reference.

**Reply: Corrected the figure reference and checked throughout the manuscript.**

**Line 317:** The DFM abbreviation is not explained.

**Reply: The DFM refers to "digital fluxgate magnetometer". Incorporated in the line 293.**

**Line 395:** Parker (1972) → Parker (1973)

**Reply: Corrected the typo error in line 456.**

**Line 608:** Reference number is missing.

**Reply: Added the reference number in the line 690.**

**Editor Comments and responses**

Thanks for adressing the reviewer comments.

Most (or all) figures have a really poor resolution, especially in the text part of the figures (e.g. Figure 1 and many more), but also for the lines, e.g. Figure 12. It looks like a problem that is arising when you export the figures from the graphics program into the word processing program or to the PDF. You have to solve this problem, before the paper can be published.

**Reply: We sincerely thank the editor for the thorough and constructive review, which has greatly contributed to improving the quality of the revised manuscript. Our detailed responses to the comments are provided in bold blue. The figure display issues in the word document have been resolved by re-exporting the figures from the graphics software. All figures have now been prepared in high-resolution TIFF format. Additionally, we have carefully reviewed the entire manuscript-including the text, figures, and formatting to ensure it is free from errors.**

Line 17: of the 3D -> of a 3D

**Reply: Revised the sentence as suggested.**

line 20: Explain ERT, EVRI in the abstract.

**Reply: Explained about the ERT and EVRI in the abstract.**

The meaning of PVR and ABS should be explained not only in the caption of Figure 1, but also in the main text.

**Reply: We explained the meaning of PVR and ABS room in the main text as suggested (line 62:63).**

Line 268: RTE needs to be explained.

**Reply: RTE stands for 'Reduced to the Equator,' already explained in lines 271:273 of the revised manuscript.**